# Clinical Benefits and Challenges in Application of Novel Portable Gastric Capsule Endoscopy for Home Healthcare Patients

**DOI:** 10.3390/diagnostics12071755

**Published:** 2022-07-20

**Authors:** Yang-Chao Lin, Ching-Lin Chen, Yi-Wei Kao, Ching-Yao Tsai, Mingchih Chen, Chih-Kuang Liu

**Affiliations:** 1Department of Gastroenterology and Hepatology, Fu Jen Catholic University Hospital, New Taipei City 242, Taiwan; yangchao.lin@gmail.com; 2Graduate Institute of Business Administration, Fu Jen Catholic University, New Taipei City 242, Taiwan; lin1006820138@gmail.com (C.-L.C.); kyw498762030@gmail.com (Y.-W.K.); ayitgroup@gmail.com (C.-Y.T.); 3Department of Internal Medicine, Zhong-Xing Branch Taipei City Hospital, Taipei City 103, Taiwan; 4Artificial Intelligence Development Center, Fu Jen Catholic University, New Taipei City 242, Taiwan; 5Institute of Public Health, National Yang Ming Chiao Tung University, Taipei City 112, Taiwan

**Keywords:** gastric capsule endoscopy, portable endoscopy, upper gastrointestinal tract, home care

## Abstract

Portable magnetic-assisted capsule endoscopy (MACE) provides satisfactory patient experience and safety with comparable performance in diagnosis of organic lesions when compared to conventional upper gastrointestinal endoscopy. In this study, a total of 58 homecare patients were included for MACE either in the hospital (*n* = 42) or at home (*n* = 16), with mean age of 71.1 ± 12.4 years. A total of 55 patients (94.83%) had completed the MACE with diagnosis of reflux esophagitis (43.6%), gastritis (54.5%), erosions (21.8%), fundic polyps (14.5%), peptic ulcers (25.9%), etc. Most patients (*n* = 47, 85.5%) were satisfied with the experience, and all patients who received MACE at home (*n* = 15, 100%) appreciated the convenience of endoscopy at home. Less than half of the patients (*n* = 24, 43.6%) could afford MACE if the expense was not covered by health insurance (USD 714). Time consumption from both traffic and capsule manipulation was also challenging for the physicians, as it took an average of 24.7 min to complete MACE, but it added up to a total of 92.7 min at home, which is about 15 times that of conventional endoscopy in hospital. More efforts are needed to ease the financial burden of patients, and optimization of workflow in community practice may help lift the obstacles revealed in this study.

## 1. Introduction

The demand for home healthcare or so-called patient-centered care is growing worldwide as people live longer. However, senior adults, especially aged more than 65 years old, are at increased risk of multiple chronic diseases and health issues, including associated functional impairment [1]. Recent advances of portable medical devices such as electrocardiogram and ultrasound equipment are being widely incorporated in community home care [2,3]. Thanks to the development of capsule endoscopy (CE), various types of capsules have been developed to detect lesions in various parts of the gastrointestinal (GI) tracts.

Portable gastric capsule endoscopy with a handheld magnetic navigator, also known as magnetic-assisted capsule endoscopy (MACE), has been an alternative for providing upper gastrointestinal examinations for in-home patients since 2020 [4]. Incorporation of MACE for home healthcare patients is important due to the huge demand for home healthcare service in an aged society. Furthermore, peptic ulcers are not uncommon in geriatric populations, and they may present as epigastralgia, bloating, easy satiety, or vomiting [5], which is also an important cause of morbidity. Mortality can occur among certain affected individuals, such as those with multiple chronic diseases; therefore, proper identification and in-time treatment of peptic ulcer disease is imperative for decreasing its associated sequelae [6,7].

Capsule endoscopy has been utilized for more than 20 years since its first launch in 2000 [8]. It provides an alternative method for investigating the gastrointestinal tracts, especially in the tubular structures such as the esophagus, small intestine, and the large bowel. Most of the commercial products in the market are designed for small intestine observations. However, the movement of capsule endoscopy relies on the gravity and peristalsis of the gastrointestinal tracts, which make active observation of mucosa in saccular organs such as the stomach impossible. To empower the capsule with controllable movement, a new device was introduced in 2009, where a capsule designed for colon examination was modified with a magnet equipped at one end of the capsule [9]. This modification made it possible for the capsule to be controlled by applying an external magnetic field. In the recent decade, application of the external magnetic field to manipulate the capsule movement, called magnetic-assisted capsule endoscopy (MACE), using either a hand-held device or robotic arms, was subsequently developed for upper gastrointestinal examination. Studies revealed that MACE has similar accuracy and completeness for observing upper gastrointestinal tract mucosa compared to conventional esophagogastroduodenoscopy (EGD) [10,11,12,13].

Portability is one of the distinguished highlights of the MACE system with a handheld magnetic navigator. This further broadens its application in examination of upper GI tracts outside the hospital, providing an innovative method of performing endoscopic examinations for home healthcare patients. Moreover, patients benefit from less discomfort, easier swallowing, anesthesia-free treatment, and no risk of cross-infection when receiving MACE compared to conventional EGD. Based on its portability and convenience, application of MACE for aged and disabled in-home patients has been carried out in Taiwan since 2020 [4], showing that portable MACE is a feasible alternative for in-home patients to receive upper GI endoscopy at home with adequate safety and accessibility and acceptable accuracy compared to that of conventional EGD in the hospital. However, some issues regarding the MACE expense, time consumption due to traffic, physician labor, and patient experience have been noted, and these factors may affect the application of portable MACE in a larger scale of population. This study analyzes the factors affecting patients’ decisions when choosing MACE or conventional EGD and challenges encountered by physicians providing MACE for in-home patients.

## 2. Materials and Methods

### 2.1. Participants

A total of 225 patients qualified and registered at “Home-based Medical Integration Program” for home healthcare were considered for the study. These patients were either senior citizens older than the age of 65 or they had certain physical disabilities and were confined to bed or needed assistance for daily activities. Registered patients with gastrointestinal complaints such as acid reflux, heartburn, nausea, vomiting, and epigastralgia were reported by community nurses for evaluation of indications for MACE. Patients with previous gastrointestinal tract obstruction or surgery, malignancy, difficulty in swallowing, pacemaker, or metal prosthetic implants were excluded from the study. A total of 58 patients received MACE. In total, 42 of 58 patients received MACE in hospital; the other 16 patients received MACE at home or in their residence. Written consent for endoscopic examinations was obtained from the patients in advance.

Patients or their caregivers were interviewed by our social workers before and after the MACE procedure. They were encouraged to express their experiences of conventional EGD and MACE, and how they felt about the MACE compared to conventional EGD. In addition, they were asked about the type of endoscopy preferred in the future after their MACE experience was addressed (Appendix A).

### 2.2. Endoscopic Equipment

We used a portable magnetic-assisted capsule endoscope system (InsightEyes^®^) developed by Insight Medical Solution (IMS) Corporation, Hsinchu City, Taiwan.

### 2.3. Data Collection

This is a retrospective study on a prospectively designed novel medical service. Patient data including gender, age, scores of activities of daily living (ADL), socioeconomic status, comorbidities, MACE reports, and interview data were collected from the Hospital Information System (HIS) with permission of Taipei City Hospital Research Ethics Committee (TCHIRB-11105015-E).

### 2.4. Data Analyses

Descriptive analysis was applied to address the possible challenges and factors related to further portable MACE utilization.

## 3. Results

Of the 58 patients, mean age was 71.1 ± 12.4 years (47 to 99 years). In total, 55 patients (94.8%) completed the MACE, with 3 patients failing to swallow the capsule after at least five attempts. There were no reported adverse events including vomiting, vigorous belching, or aspiration during or after the examinations. The average endoscopic examination time was 24.7± 8.1 min (11 to 44 min) for each patient. The endoscopic results yielded an average of 1.6 organic lesions for each examined patient in the upper gastrointestinal tracts. The corresponding prevalence of lesions among the patients was: reflux esophagitis (43.6%), gastritis (54.5%), erosions (21.8%), fundic polyps (14.5%), gastric ulcers (14.5%), duodenal ulcers (12.7%), and adenomatous change of ampulla of Vater (1.8%) (Figure 1). All patients were treated or referred for further examinations accordingly.

We further divided the patients into two groups for analysis: MACE at hospital and MACE at home. There were 42 patients in the hospital group and 16 patients in the home group. Procedure time and organic lesions discovered by MACE were listed in Table 1.

In the home group, mean age was 74.3 ± 15.4 years (47 to 99 years); 15 of the 16 patients (93.8%) completed the MACE except one 83-year-old female with recent gastric ulcer. The average endoscopic examination time was 23.7 ± 10.0 min (14.1 to 42.5 min) for each person. It yielded an average of 2.4 organic lesions for each examined patient in the in-home group (Table 1); the corresponding prevalence of lesions among the patients was: reflux esophagitis (66.7%), gastritis or erosions (46.7%), fundic polyps (13.3%), gastric ulcers (13.3%), duodenal ulcers (13.3%), and adenomatous change of ampulla of Vater (6.7%). In the hospital group, mean age was 69.6 ± 10.9 years with two patients failing to swallow the capsule. The overall success rate was 95.2%, and procedure time was 25.1 ± 7.4 min. Organic lesions in the hospital group were: GERD (35%), gastric ulcer (15%), gastric erosion (12.5%), gastritis (45%), duodenal ulcer (12.5%), and fundic gland polyps (15%). The prevalence of gastritis (80% vs. 45%, *p* = 0.045) and gastric erosion (46.7% vs. 16.5%, *p* = 0.019) was much higher in the home group.

After completing the MACE procedure, 47 (85.5%) patients expressed their preference for MACE due to the comfort of the procedure compared to that of conventional EGD. All patients were willing to receive MACE when informed that the cost of MACE (USD 714) can be reimbursed by national health insurance. In addition, 24 patients (43.6%) would choose to receive MACE in the future if necessary, even if the MACE procedure is not reimbursed by the insurance (Table 2).

In the home group, with patients in this group unable to visit or be transported to the hospital, the medical team performed MACE at patients’ home or residence. In our study, the mean distance from hospital to patients’ residence was 3.13 km (ranging from 2.2 km to 3.8 km). The average round-trip traffic time between the hospital and the community was 36.9 min (ranging from 26 to 57 min). After adding up the travel time, it took an average of 92.7 min (ranging from 72 to 130 min) to carry out each MACE for in-home patients in the community (Table 3).

## 4. Discussion

This study evaluates the feasibility of providing endoscopic examination using a novel portable MACE. Overall, MACE produced satisfactory results for use both at home and in the hospital. The overall success rate for completing the examination of the in-home group was 93.75%, which is comparable to that of the in-hospital group of 95.23%. There were no reported adverse events such as vomiting, bleeding, or severe abdominal pain during or after the MACE. The portable MACE system developed by Insight Medical Solutions (IMS) comprised a hand-held magnetic field navigator (MFN), an image processing unit, and a 21-inch high-definition monitor. All the equipment can be organized and stored in a 29-inch aluminum roller suitcase. The compactness and portability of the IMS MACE system made it more appropriate for home use than other contemporary magnetic capsules controlled by a robotic arm [14,15,16] or an electromagnetic coil system [10,17,18].

This study also showed that in-home patients with upper GI symptoms warranted consideration for endoscopic examinations, especially after careful selection of the candidates according to symptoms and comorbidities. The incidence rate of uncomplicated peptic ulcer disease among the general population was about 1 case per 1000 person-years [19]. However, the prevalence rate of endoscopic documented peptic ulcer disease ranged from 9.6% to 17.2% as noted in certain Asian studies [20,21]. In our study, peptic ulcer disease was noted in 15 (27.2%) patients with symptoms of bloating, dyspepsia, or epigastralgia, and they were all treated with proton pump inhibitors for 2 months and tested with the C-13 urea breath test to confirm *Helicobacter pylori* infection. Gastroesophageal reflux disease (GERD) was another common and chronic bothersome disorder. GERD prevalence was about 25% in the Taiwan community [22]; however, nearly half (43.6%) of our patients that had symptoms of heartburn, flatulence, or bloating had endoscopic diagnosis of reflux esophagitis, and they were given proton pump inhibitors and proper diet instructions after endoscopic diagnosis. These diagnostic results may reflect underdiagnosis of upper gastrointestinal tract disorders among the senior, multi-comorbid home healthcare population. Valuable endoscopic diagnosis made by the portable MACE guided accurate and finite treatments.

For the 16 patients in our in-home study, performing MACE at home lifted the barriers for these patients who were unable to visit or be brought to the hospital for endoscopic examinations. These senior patients may be physically disabled or confined to bed due to multiple morbidities. However, some factors that can affect further widening application of MACE for home-based patient care were noted. It is time-consuming and labor-demanding for a professional endoscopist to carry out MACE examinations outside the hospital, and this may be a major obstacle for physicians to provide this kind of community medical service without additional incentive. By the definition of the World Health Organization, there has been an aged society in Taiwan since 2018, when citizens older than 65 years exceeded 14% of the total population. The huge impact of medical demand was especially encountered in the fields of home healthcare (HHC) for the elderly and disabled, and the Ministry of Health and Welfare of Taiwan established a social-insurance-based long-term care system in 2016 [23,24] and tried to recruit more medical and para-medical staff as well as more social workers into the field of home healthcare to meet the increasing demand for manpower for HHC. However, the number of professional personnel and social workers increased more slowly than anticipated. In our study, it took an average of 23.7 min to complete a MACE procedure from capsule intake to capsule removal, but the travel time between the hospital and the patient’s residence took another 26 to 57 min. This commute made the procedure time-consuming, and the equipment setup time added another 10 min for each patient.

The high labor demand and time consumption aspects of MACE are major obstacles for physicians to participate in in-home MACE services. Several countries have introduced nurse endoscopists as a means of easing the burden of manpower demand for certain endoscopic procedures [25]. Some studies showed that nurses can provide an accurate general diagnostic upper gastrointestinal endoscopy service as competently as doctors [26,27,28]; however, non-physician endoscopy is not eligible in Taiwan at present. In addition to labor shortage, it took an average of 92.7 min to complete each MACE procedure from the hospital to the community. Travel time was an issue, especially in the metropolitan areas during rush hours. It only takes about 6.2 min (range, 2–18 min) for a physician to complete a qualified routine EGD examination in hospital [29]. Some studies even showed that a 3 min endoscopy was adequate to detect upper GI neoplasms [30]. Time consumption (92.7 min vs. 6.2 min) of MACE is a challenging issue at present.

Interestingly, patients benefited most from easy access to endoscopic examination at home, and MACE fulfills the niche that conventional EGD could not provide outside the hospital. However, the cost of time and labor demand for the medical staff can discourage physicians from further participation in the home medical services as mentioned above.

According to our study, 25.5% of the patients had “low-income household” status, and 56.4% of the patients could not afford or were unwilling to pay for a MACE examination if the expense was not covered by the national health insurance. Compared to conventional EGD with intravenous analgesia, the cost of MACE was about 5.7 times higher (USD 714 vs. USD 125). The benefits of convenience and accessibility of the MACE were outweighed by the financial burdens for nearly half of the patients. On the other hand, capsule endoscopy is incapable of performing tissue biopsies or endoscopic treatments, which may be another drawback in the clinical setting.

The capability of MACE to detect upper gastrointestinal tract lesions accurately has been a major quality concern since its debut. Several studies have addressed this issue based on different capsule models [11,12,13,18]; however, no head-to-head or direct comparison study has been conducted for conventional EGD and InsightEyes^®^ capsule endoscopy system. The completeness of observing upper GI tracts was 93.75% for esophagus, 81.25% for stomach, and 75.00% for the duodenum in a previous small study for a similar population [4].

Timely diagnosis of upper gastrointestinal lesions for in-home patients is especially valuable in not only relieving symptoms but preventing disease progression; in this study, there were 24 patients with erosive esophagitis, 15 with peptic ulcer disease, and 1 with adenoma in ampulla of Vater. All were treated accordingly with good results.

Compared to conventional endoscopy, capsule endoscopy harbors several advantages in terms of easy access, no risk of cross-infection, less invasiveness, no need for intravenous analgesia …etc., but certain weak points remain to be resolved, for instance, the high cost and the inability to perform tissue acquisition and endoscopic therapy. The practical role of MACE might be broader if the cost goes down to a point where the expense could be covered by health insurance, or the capsule could be equipped with more sensory or interventional functions.

This study has some limitations. It was a retrospective study on a small number of patients. Bias in the diagnostic results may exist and can only be applied to a certain population. It is difficult to compare diagnostic outcomes of both in-home and at-hospital groups, due to non-randomization and the heterogeneous character of the study population. Further studies in prospective settings on a large number of participants can yield more conclusive diagnostic results.

## 5. Conclusions

Magnetic-assisted capsule endoscopy is a feasible alternative to conventional endoscopy for the diagnosis of upper gastrointestinal disorders, both in hospital and at home. Endoscopic examinations were made possible for in-home patients by application of the portable MACE. MACE showed satisfactory results with high success rates, satisfying patient experience, and easy access to endoscopic examination, and it generated valuable endoscopic diagnosis that helped to treat the patients with proper medication. However, in-house MACE procedures were both time-consuming and labor-demanding for the physicians, and the high expense of the procedure (if not covered by insurance) for the patients can be a major obstacle for broader application in the near future.

## Figures and Tables

**Figure 1 diagnostics-12-01755-f001:**
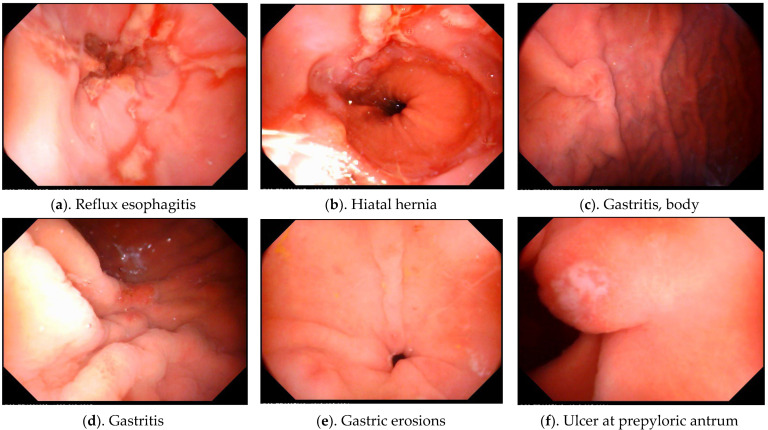
Organic lesions discovered in home healthcare patients using portable MACE. (**a**) Reflux esophagitis, (**b**) hiatus hernia, (**c**,**d**) gastritis, (**e**) gastric erosions, (**f**) ulcer, (**g**) fundic gland polyp, (**h**) duodenal ulcer, (**i**) ampulla of Vater. Note: Pictures (**g**,**i**) were adapted from a previously published article with an overlapping study population with permission adapted from Ref. [4]. 2021, Lin, Y.-C.

**Table 1 diagnostics-12-01755-t001:** Patient characteristics and organic lesions discovered by MACE.

	Total MACE	MACE at Home	MACE at Hospital	*p*
MACE enrollment (*n*)	58	16	42	
Success MACE (%)	55 (94.8%)	15 (93.8%)	40 (95.2%)	1.001
Age (mean (SD))	71.1 (12.4)	73.73 (15.8)	69.55 (10.9)	0.270
Female	28	7 (46.7)	21 (52.5)	
Male	27	8 (53.3)	19 (47.5)	
Procedure time (mean (SD))	24.7 (8.1)	23.67 (10.0)	25.07 (7.4)	0.571
Organic lesions (*n* (%))	90	36	54	
Gastroesophageal reflux disease	24 (43.6)	10 (66.7)	14 (35.0)	0.072
Gastric ulcer	8 (14.5)	2 (13.3)	6 (15.0)	1.001
Duodenal ulcer	7 (12.7)	2 (13.3)	5 (12.5)	1.001
Gastritis	30 (54.5)	12 (80.0)	18 (45.0)	0.045
Gastric erosion	12 (21.8)	7 (46.7)	5 (12.5)	0.019
Fundic gland polyp	8 (14.5)	2 (13.3)	6 (15.0)	1.001
Ampulla of Vater adenoma	1 (1.8)	1 (6.7)	0 (0.0)	0.607

**Table 2 diagnostics-12-01755-t002:** Patient experience and conditional preference for MACE.

Group	Post-MACE Experience Compared to Conventional EGD		Preference for MACE if Expense		Low-Income Household
	More Comfortable	Easy Access at Home	Covered by Insurance	Not Covered by Insurance	
Hospital	35 (87.5%)		40 (100%)	19 (47.5%)	9 (22.5%)
Home	12 (80.0%)	15 (100%)	15 (100%)	5 (33.3%)	5 (33.3%)
Total	47 (85.5%)		55 (100%)	24 (43.6%)	14 (25.5%)

**Table 3 diagnostics-12-01755-t003:** In-home group patient characteristics, home distance, and time needed to carry out MACE services. Note: Part of the data (procedure time) with permission adapted from Ref. [4]. 2021, Lin, Y.-C.

Case	Age	Sex	Period of Home Care (Month)	Previous Conventional EGD	Round Trip Distance between Hospital and Residence (Km)	Round Trip Traffic Time (min)	Procedure Time (min)	Time Expenditure per Service (min)
1	63	M	12	Yes	4.8	28	30	88
2	70	M	36	Yes	5.2	29	34	95
3	77	F	60	Yes	6.0	28	18	81
4	70	M	30	Yes	7.6	57	42	130
5	56	M	24	Yes	5.0	36	11	78
6	82	F	120	Yes	5.0	33	24	95
7	91	M	32	Yes	6.0	40	19	79
8	90	M	36	Yes	6.8	42	14	90
9	47	F	84	Yes	7.2	50	23	110
10	69	F	46	Yes	7.6	55	19	106
11	90	F	4	Yes	4.6	28	41	100
12	87	M	12	Yes	5.2	30	15	77
13	53	M	7	Yes	4.4	26	18	75
14	62	F	9	Yes	6.0	42	33	108
15	99	F	38	Yes	4.8	30	14	79
16	83	F	11	Yes	5.3 (not counted)	33 (not counted)	NA	NA
Average (±SD)			35.1 ± 31.2	68.75%	5.75 ± 1.10	36.9 ± 10.3	23.7 ± 10.0	92.7 ± 12.8

## Data Availability

Not applicable.

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
