# Peer review of "Clinical Benefits and Challenges in Application of Novel Portable Gastric Capsule Endoscopy for Home Healthcare Patients"

_diagnostics, 2022, doi:10.3390/diagnostics12071755_

Round 1

Reviewer 1 Report

The authors describe the results of new equipment which makes possible at-home gastric endoscopy by a guided capsule. The report is clearly descriptive and evaluates the difficulties and benefits of this procedure.  The authors consider the self-report of patients related to comfort and costs of this method, additionally, they evaluated the accuracy of the procedure when compared with conventional hospital endoscopy results. The aspects evaluated are relevant and up-to-date. 

Author Response

Thank you very much for your appreciation and comments.

Reviewer 2 Report

I’ve read with great interest the paper on “Innovations and Challenges in Application of Portable Gastric Capsule Endoscopy for Home Healthcare Patients”, which describes the feasibility and diagnostic yield of MACE in patients with at home health care.
The principle is innovative and useful to detect upper gastrointestinal pathology in this category of patients, who may otherwise present with complications.
The title is too vague and does not accurately reflect the study design and aim. I would suggest rephrasing.

The introduction is well written, providing the reader with background on the use of capsule endoscopy and its utility in patients who cannot undergo endoscopy, which is the case of the population studied in this study.

Methods are accurately described and results are relevant.

There is a very small discussion on cost-efficacy of this method. Given the high costs of MACR compared to conventional endoscopy, there should be some data about rate of lesion detectionn(using some comparator), also about implications for management (need for prescriptions, or other means of therapy, also impact on progression to complications). Also, there should be a more focused discussion regarding noninvasiveness of the procedure compared to endoscopy. On the other hand, endoscopy would be favoured in patients with pacemakers, which may be the case in ederly patients with at home care.

In summary a good paper regarding the proof of concept - MACE in persons with at home healthcare - but more extensive discussion is needed about its practical role and advantges /disadvantages compared to enodoscopy.

Author Response

Dear reviewer:

Thank you very much for your valulable comments, please see the attchments for my replies.

Reviewer 3 Report

This reported that the advantage of portable magnetic assisted capsule endoscopy (MACE) for in-home patients. MACE is less invasive. Portable use will broaden the indication of upper gastrointestinal examination. I have some comments.

The data of in-home patients are same of Ref.4(Healthcare (Basel). 2021 May 13;9(5):577.). Some pictures are also from same paper. It should be described.

  Comparisons between in-home group and at-hospital group are difficult because the selection criteria for patients in the hospital are not randomized.

Author Response

Dear Reviewer:

Thank you very much for your valuable comments, please refer to the attachment for my replies. 

Round 2

Reviewer 3 Report

This reported that the advantage of portable magnetic assisted capsule endoscopy (MACE) for in-home patients. MACE is less invasive. I have no further comments on this submission.